# Treatment outcomes of patients with MDR-TB and its determinants at referral hospitals in Ethiopia

**Mengistu K. Wakjira** [1]*, **Peter T. Sandy**[2], **A. H. Mavhandu-Mudzusi**[3]

**1** Abt Associates Inc Ethiopia, Addis Ababa, Ethiopia, **2** Buckinghamshire New University, Uxbridge Campus, Uxbridge, London, United Kingdom, **3** University of South Africa, College of Human Sciences, Office of Graduate Studies and Research, Pretoria, South Africa

\* mengistukenea@gmail.com

## Abstract

### Background

There is limited empirical evidence in Ethiopia on the determinants of treatment outcomes of patients with multidrug-resistant tuberculosis (MDR-TB) who were enrolled to second-line anti-tuberculosis drugs. Thus, this study investigated the determinants of treatment outcomes in patients with MDR-TB at referral hospitals in Ethiopia.

### Design and methods

This study was underpinned by a cross-sectional quantitative research design that guided both data collection and analysis. Data is collected using structured questionnaire and data analyses was performed using the Statistical Package for Social Sciences. Multi-variable logistic regression was used to control for confounders in determining the association between treatment outcomes of patients with MDR-TB and selected predictor variables, such as co-morbidity with MDR-TB and body mass index.

### Results

From the total of 136 patients with MDR-TB included in this study, 31% had some co-morbidity with MDR-TB at baseline, and 64% of the patients had a body mass index of less than 18.5 kg/m$^2$. At 24 months after commencing treatment, 76 (69%), n = 110), of the patients had successfully completed treatment, while 30 (27%) died of the disease. The odds of death was significantly higher among patients with low body mass index (AOR = 2.734, 95% CI: 1.01–7.395; P<0.048) and those with some co-morbidity at baseline (AOR = 4.260, 95% CI: 1.607–11.29; p<0.004).

### Conclusion

The higher proportion of mortality among patients treated for MDR-TB at Adama and Nekemte Hospitals, central Ethiopia, is attributable to co-morbidities with MDR-TB, including HIV/AIDS and malnutrition. Improving socio-economic and nutritional support and provision

**Data Availability Statement:** All relevant data are within the paper and its Supporting Information files.

**Funding:** During the time the research was conducted MKW was employed to Abt Associates Inc. operating in Ethiopia. The funder provided support in the form of salaries for authors (MKW) but did not have any additional role in the study design, data collection and analysis, decision to publish, or preparation of the manuscript. The specific roles of these authors are articulated in the 'authors contributions section.

**Competing interests:** During the time the research was conducted MKW was employed to Abt Associates Inc. operating in Ethiopia. There are no patents, products in development or marketed products associated with this research to declare. But the authors hereby confirm that this does not alter our adherence to PLOSONE policies on sharing data and materials.

of integrated care for MDR-TB and HIV/AIDS is recommended to mitigate the higher level of death among patients treated for MDR-TB.

# Background

During the past two centuries, tuberculosis (TB) has claimed more lives than any other infectious disease on earth [1]. Despite advances in the treatment, prevention and control of the disease, tuberculosis continues to be one of the major causes of morbidity and mortality worldwide [2–5]. This is attributed mainly to the global rise in the incidence of MDR-TB [6–8]. MDR-TB limits treatment options for the disease. Treatment of MDR-TB can take up to two years with drugs that are poorly tolerated and difficult to monitor. Moreover, the treatment outcomes of patients with MDR-TB are generally poor [9–17]. Several factors have an impact on the outcomes of the treatment given for MDR-TB. Social and financial hardships, including undernutrition, enhance vulnerability to tuberculosis and challenge the process and the outcomes of the treatment given for the disease [18–22]. Co-morbidities with MDR-TB, including HIV, malignancies, diabetes mellitus and chronic renal failure are risk factors for death among patients with MDR-TB [23–31]. Moreover, adverse drug reactions from second-line drugs and MDR-TB associated stigma have a negative impact on the outcomes of the treatment given for MDR-TB [32, 33]. Without addressing these factors, the mere provision of free drugs may not directly lead to optimum treatment outcomes of patients treated for the disease [34, 35].

Ethiopia is one of the global high-burden countries for TB, TB/HIV and MDR-TB [16]. Yet, the number of MDR-TB cases detected and enrolled for treatment in the country has been far below the national incident estimate [17]. The huge pool of untreated cases of MDR-TB represents an important source of disease transmission [19]. Thus, the government of Ethiopia is expanding the services on the programmatic management of MDR-TB to all its regions using community-based ambulatory model of care [20].

In Ethiopia, there is very limited empirical evidence on factors determining the outcomes of the treatment given for patients with MDR-TB based on the community-based ambulatory model of care. Absence of such empirical evidence is a challenge to understand the desired impact of the national MDR-TB service expansion. Thus, we undertook a study to assess the determinants of the treatment outcomes of patients with MDR-TB at two referral hospitals found in the Oromia region of Ethiopia.

## Aim and objectives

This study aimed to investigate the treatment outcomes of patients with MDR-TB and its determinants at referral hospitals in the Oromia region of Ethiopia. The specific objectives of the study were to:

- determine the treatment outcomes of patients with MDR-TB who are enrolled on second-line anti-tuberculosis drugs.

- assess factors associated with observed levels of treatment outcomes among patients treated for MDR-TB.

# Materials and methods

## Study design

The study utilised a cross-sectional quantitative research design to assess, describe and analyse the treatment outcomes of patients with MDR-TB, and its determinants. Thus, a deductive

approach was used to test the plausible relationship between the treatment outcomes of patients with MDR-TB and its determinants.

## Study setting and sampling

The study was conducted at Adama and Nekemte Referral Hospitals in the Oromia region of Ethiopia. In the Oromia region of Ethiopia, the programmatic management of MDR-TB was first initiated at these hospitals. Thus, it was assumed that the two hospitals had adequate experience and data on the programmatic management of MDR-TB. The population of the study consisted of all patients with laboratory-confirmed MDR/RR-TB enrolled for the treatment for MDR-TB at the two hospitals. Between 26 December 2012 and 17 September 2016, a total of 182 patients with MDR-TB were enrolled for treatment at the two hospitals. Data is collected from all patients who were on treatment for a period of six months or above at the time of data collection. 46 (25%) patients did not meet the inclusion criteria (less than 6 months on treatment by the time of data collection and for whom at least interim treatment outcome could not be determined) and excluded from the study. Thus, data is collected from the remaining 136 (75%) patients that fulfilled the inclusion criteria. All patients that met the inclusion criteria were included in the study to make sure that sufficient number of the required sample size is included in the study and inferences could be made. The main variable of interest among the clinical characteristics of patients with MDR-TB was the presence of co-morbidity with MDR-TB. There are two groups with respect to this: group of patients with co-morbidity (p1) and groups without co-morbidity (p2) with MDR-TB at the baseline. From the literature P1 & P2 for this study are, P1 = 0.81 and p2 = 0.70 [39]. The calculated power for the sample size included in this study is 0.87. Thus, clinical record of all the 136 participants who meet the inclusion criteria is retrieved and included in the study.

## Data collection and analysis

Ethical approval to conduct the study was obtained from the Department of Health Studies of the University of South Africa (UNISA) and from Oromia Region Health Bureau, Department of Public Health Emergency Management and Health Research. To get access to patients' medical records at each study site, informed consent was sought and obtained from hospital management and clinical caregivers [36]. The study was conducted between 10 November 2016 and 7 February 2017, by using a structured questionnaire. Study variables were extracted from extant literature and conceptualised and operationalised in line with the aims and objectives of the study. The questionnaire was serially reviewed by experienced experts and subjected to preliminary field testing and further amendment before its use on the main study. Data was collected by the principal investigator and two healthcare professionals who were offered two days' training on the specifics of data collection for this study. Data is analysed using Statistical Package for Social Sciences [37]. Categorical variables were summarised as frequencies and percentages. A bivariate analysis was performed to identify factors associated with the treatment outcome of patients with MDR-TB. A multi-variable logistic regression analysis was employed to determine the independent predictors of the treatment outcomes of patients with MDR-TB. The results of the logistic regression are expressed as crude and adjusted odds ratio. Confidence intervals and p-values were used to test significance of the observed sample parameters in exploring determinants of the treatment outcomes of patients treated for MDR-TB.

## Results

### General characteristics of the participants

A total of 136 patients (n = 136) with MDR-TB were included in the study. 74 (54%) were male and 62 (46%) were female patients with MDR-TB. Altogether, 128 (94%) of the patients were in the productive age group of 15 to 64 years. Moreover, 28 out of 30 (93%) of the total deaths from MDR-TB occurred in the same age group. 4 (3%) of the patients were aged less than 15 years, while 4 (3%) were aged 65 years and above. The mean age of the study participants (mean ± SD) was 32.12 ± 12.53 while the age range of the participants was 4 to 73 years. From those within age of employment (n = 132), 70 (53%) were self-employed mainly in the informal labour work while 46 (35%) were not employed (Table 1).

### Clinical characteristics of the participants

Altogether 134 (98%) of the patients were bacteriologically confirmed pulmonary MDR-TB cases, while 2 (2%) were bacteriologically confirmed extra-pulmonary MDR-TB. 90 (66%) of the patients were diagnosed with MDR-TB after failure of re-treatment regimen with first-line anti-TB drugs, while 17 (13%) were diagnosed after failure of the standard six-month regimen with first-line anti-tuberculosis drugs. A total of 14 (10%) patients were registered for treatment after relapse, while 11 (8%) were new cases of MDR-TB without any prior history of treatment with anti-tuberculosis drugs. 4 (3%) of the patients were diagnosed among patients returning after lost to follow-ups. From patients with documented baseline sputum microscopy test (n = 132), 105 (79%) were sputum smear positive and 27 (21%) were sputum smear negative. An analysis of initial bacillary load (n = 132) revealed that the initial bacillary load for 59 (45%) patients was scanty, and it was moderate for 41 (31%) and high for 5 (4%) patients. Altogether 89 (65%) of the total patients had a drug-susceptibility test result by GeneXpert and they were resistant to rifampicin. Only 47 (35%) of the total patients were diagnosed using either culture or line probe assay tests. This group of patients had drug-susceptibility test result for both rifampicin and isoniazid, and they were resistant to both drugs. No patient had a drug-susceptibility test result for any of the second-line anti-tuberculosis drugs. A total of 41 (31%) patients (n = 133) had some co-morbidity with MDR-TB at baseline, out of which 34 (83%) was due to co-infection with HIV. 5 (12%) patients had diabetes mellitus and each of 2 (5%) patients had cardio-vascular and kidney diseases. 87 (64%) of the 136 patients had body mass index (BMI) of less than 18.5kg/m$^2$ at baseline, which is indicative of malnutrition as co-morbidity with MDR-TB (Table 1). Complete data on adverse drug reactions from second-line anti-TB drugs was retrieved only for 91 (67%) (n = 91), of patients included in the study, and all these patients experienced at least one episode of adverse drug reactions in the course of their treatment. A total of 31 (34%) patients experienced five or more episodes of adverse drug reactions, 12 (13%) experienced four episodes, 14 (15%) experienced three episodes, 22 (24%) experienced two episodes, while the remaining 12 (13%) experienced one episode of adverse drug reactions. By the body organs involved, 73 (80.2%) patients developed gastro-intestinal tract-related adverse reactions and 35 (38.5%) patients had neurological-related adverse drug reactions. Musculoskeletal-related adverse reactions occurred among 26 (28.6%), followed by cardio-vascular-related adverse drug reactions that occurred in 24 (26.4%) patients. Electrolyte disturbances occurred among 13 (14.3%) patients and 11 (12%) developed psychiatric-related adverse drug reactions. Moreover, 5 (5.5%) of the patients developed adverse drug reaction involving the eye while 3 (3.2%) developed immune related adverse drug reactions. Permanent loss of hearing occurred in 7 (7.7%) patients and 1 (1%) patient died of suicide while on treatment. Analysis of trend of occurrence of the adverse drug-

**Table 1. Socio-demographic and clinical characteristic of study participants (n = 136).**

| Parameter | n (%) |
|---|---|
| **Sex (n = 136)** | |
| Male | 73 (54) |
| Female | 63 (46) |
| **Age category (n = 136)** | |
| <15 years | 4 (3) |
| 15–44 years | 110 (81) |
| 45–64 years | 18 (13) |
| >/ = 65 years | 4 (3) |
| **Patients' employment status (n = 132)** | |
| Formally employed | 7 (5.3) |
| Self-employed | 70 (53) |
| Unemployed | 46 (35) |
| Other | 9 (7) |
| **Patients' drug-resistance type at diagnosis (n = 136)** | |
| RR | 89 (65) |
| MDR-TB | 47 (35) |
| **HIV test result (n = 131)** | |
| HIV positive | 34(26) |
| HIV negative | 97 (74) |
| **Presence of co-morbidity at baseline (n = 133)** | |
| Yes | 41 (31) |
| No | 92 (69) |
| **Type of co-morbidity at baseline (n = 41)** | |
| HIV/AIDS | 34 (83) |
| Diabetes mellitus | 5 (12) |
| Other | 2 (5) |
| **Body mass index (BMI) (n = 136)** | |
| BMI <18.5Kg/m$^2$ | 87 (64) |
| BMI >/ = 18.5Kg/m$^2$ | 49 (36) |
| **Site of the TB disease (n = 136)** | |
| Pulmonary | 134 (98) |
| Extra-pulmonary | 2 (2) |
| **Type of the TB case (n = 136)** | |
| Bacteriologically confirmed PTB | 134 (98) |
| Bacteriologically confirmed EPTB | 1 (1) |
| Clinically diagnosed EPTB | 1 (1) |
| **Result of diagnostic sputum smear examination (n = 132)** | |
| Smear positive | 105 (79) |
| Smear negative | 27 (21) |
| **Sputum bacillary load reported at diagnosis (n = 132)** | |
| No AFB seen | 27(20) |
| Scanty | 59(45) |
| Moderate | 41(31) |
| High | 5(4) |

n = number; % = percent.

reactions revealed that except for musculo-skeletal and neurological-related adverse drug reactions, most of the adverse drug reactions occurred during the first six to eight intensive-phase months of patient treatment (Table 1).

## Treatment outcomes of patients with MDR-TB

At six months of follow-ups, 98 (72%) of the patients were culture negative and 26 (19%) died of the disease while the six month's interim treatment outcome was not evaluated and documented for 12 (9%) of the patients (Table 2).

The 24 months' treatment outcome was determined for 110 (81%) of the patients. At 24 months, 76 (69%) had successfully completed their treatment. Thus, the composite treatment success rate for patients included in this study was 69%. Altogether 30 (27%) patients died of the disease at the 24 months' follow up period. The treatment outcomes of 3 (3%) patients were not evaluated at month 24, mainly due to patient transfers to other treatment centres and reports on their treatment outcomes were not available by the time of data collection. 1 (1%) patient was lost to follow-ups and not retrieved by the time of data collection.

## Determinants of the treatment outcomes of patients with MDR-TB

Bivariate analysis revealed that from the total of 30 deaths that occurred among all patients, 19 (42%) was among female patients, while 11 (23%) was among male patients. Compared to male patients with MDR-TB, the odds of death was higher among female patients with MDR-TB (Crude OR = 2.436; $X^2$ = 4.459; P<0.035; 95%CI = 1.066–5.566). Compared to patients without any co-morbidity at the baseline, the odds of death was higher among patients with some co-morbidity with MDR-TB at the baseline (Crude OR = 2.864; $X^2$ = 5.802; P<0.016; 95%CI = 1.217–6.743). Likewise, compared to HIV-negative patients with MDR-TB, the odds of death was higher among patients co-infected with HIV (Crude OR = 2.741; $X^2$ = 4.795; P<0.029; 95%CI = 1.112–6.761). Moreover, compared to patients whose body mass index (BMI) was greater than or equal to 18.5kg/m$^2$, the odds of death from MDR-TB was higher among patients with BMI <18.5kg/m$^2$ (Crude OR = 2.925; $X^2$ = 5.327; P<0.021; 95% CI = 1.176–7.277) (Table 3).

Furthermore, the presence of fibrotic (extensive) lung lesion, which is indicative of advanced disease status, was revealed to be predictor of death from MDR-TB (Phi $X^2$ = 0.405, P<0.017).

The final multivariable logistic regression analysis (Table 4) revealed that the odds of death among patients with MDR-TB who had some co-morbidity with MDR-TB at the baseline was

**Table 2. The treatment outcomes of patients with MDR-TB at six and 24 months after commencing treatment.**

| Six-month treatment outcomes (n = 136) | | 24-month (final) treatment outcomes (n = 110) | |
|---|---|---|---|
| Parameter | n (%) | Parameter | n (%) |
| Culture negative | 98 (72) | Cured | 65 (59) |
| Culture positive | 0 (0) | Treatment completed | 11 (10) |
| Died by sixth month | 26 (19) | Composite treatment success rate | 76 (69) |
| | | Died by 24 months | 30 (27) |
| Not evaluated | 12 (9) | Lost to follow-ups | 1 (1) |
| Lost to follow-ups | 0 (0) | Not evaluated | 3 (3) |
| | | On treatment | 26 (19) |

n = number; % = percent.

**Table 3. Summary of bivariate analysis on the determinants of MDR-TB treatment outcomes of the study participants at Adama and Nekemte Hospitals, Oromia, Ethiopia, December 2012 –September 2016 (n = 110).**

| Variable | Category | Favourable treatment outcome (cured or treatment completed) n (%) | Unfavourable treatment outcome (Death) n (%) | Total n (%) | Crude OR[@] | Wald $x^2$ test result | P-value | 95% CI[@@] |
|---|---|---|---|---|---|---|---|---|
| Sex | Male | 54 (77) | 11 (23) | 65 (59) | ___ | ___ | ___ | ___ |
| | Female | 26 (58) | 19 (42) | 45 (41) | 2.436 | 4.459 | <0.035 | 1.066–5.566 |
| BMI | >18.5Kg/m$^2$ | 36 (82) | 8 (18) | 44 (40) | ___ | ___ | ___ | ___ |
| | </ = 18.5Kg/m$^2$ | 40 (60) | 26 (39) | 66 (60) | 2.925 | 5.327 | <0.021 | 1.176–7.277 |
| Any co-morbidity at baseline | No | 58 (76) | 18 (24) | 76 (69) | ___ | ___ | ___ | ___ |
| | Yes | 18 (53) | 16 (47) | 34 (31) | 2.864 | 5.802 | <0.016 | 1.217–6.743 |
| HIV | Negative | 62 (75) | 21 (25) | 83 (75) | ___ | ___ | ___ | ___ |
| | Positive | 14 (52) | 13 (48) | 27 (25) | 2.741 | 4.795 | <0.029 | 1.112–6.761 |
| Resistance type | RR-TB | 39 (61) | 25 (39) | 64 (58) | ___ | ___ | ___ | ___ |
| | MDR-TB | 37 (80) | 9 (20) | 46 (42) | 2.635 | 4.608 | <0.032 | 1.088–6.384 |

n = number; % = percent

@ = Odds Ratio

@@ = Confidence Interval.

significantly higher than the odds of death among those without co-morbidity (AOR = 4.260, 95%CI: 1.607–11.297; P<0.004). Moreover, the odds of death from MDR-TB among patients with low body mass index (MBI) was 2.7 times higher than the odds of death among patients with body mass index greater than or equal to 18.5kg/m$^2$ (AOR = 2.734, 95%CI: 1.01–7.395; P<0.048). Likewise, the odds of death from MDR-TB among female patients with MDR-TB was significantly higher than the odds of death among male patients with MDR-TB (AOR = 2.511, 95%CI: 1.005–6.272; P<0.049). In summary, about 26% of the total deaths from MDR-TB, observed in this study, was explained by the presence of some co-morbidity with MDR-TB at the baseline, and being a female patient with MDR-TB (Nagelkerke R Square = 0.257).

**Table 4. Results of the multivariable analysis using logistic regression on factors associated with unfavourable MDR-TB treatment outcome of the study participants at Adama and Nekemte referral hospitals, Oromia, Ethiopia, December 2012 –September 2016.**

| Variable | Crude OR[@] | 95% CI[@@] | P-value | Adjusted OR | 95% CI | P-value |
|---|---|---|---|---|---|---|
| Presence of any co-morbidity at the baseline | 2.864 | 1.217–6.743 | 0.016 | 4.260 | 1.607–11.297 | 0.004 |
| Body mass index (BMI) | 2.925 | 1.176–7.277 | 0.021 | 2.734 | 1.01–7.395 | 0.048 |
| Sex | 2.436 | 1.066–5.566 | 0.035 | 2.511 | 1.005–6.272 | 0.049 |
| HIV | 2.7.41 | 1.112–6.761 | 0.029 | 0.088 | | 0.767 |
| Drug-resistance type | 2.635 | 1.088–6.384 | 0.032 | 2.630 | | 0.105 |

@ = Odds Ratio;

@@ = Confidence Interval.

### Respiratory MDR-TB infection control practices at the study sites

From the total of 114 (n = 114) patients who lived with at least one househld contact, tracing and evaluation of contacts was conducted only for 60 (53%) patients. For the remaining 54 (47%) patients, it was unknown whether any of their household contacts were traced. Assessment of respiratory MDR-TB infection control practices during patients' movement between the hospitals and community level treatment follow-up centres showed that, during the initial patient linkage to the community 97 (92%) (n = 105) of the patients were transported using hospital ambulances and were also escorted by the nurse caregivers at the hospitals. After initial patient linkage to the community, patients used public transport services during their travels to hospitals to attend the monthly MDR-TB follow-up services. Moreover, patients from the remote rural areas use public transport to attend treatment at the community treatment follow-up centres. Analysis of respiratory MDR-TB infection control practice at the patients' household level revealed that, for 105 (77%) patients who live with family, caregivers did not visit the patients' homes to educate patient families on the risk of respiratory MDR-TB transmission and make housing arrangement before patients are sent back home. This study revealed that 8 (6%) of the total patients included in the study were diagnosed among household contacts of index patients with MDR-TB.

## Discussions

At six months after commencing the treatment for MDR-TB, 72% of the patients were culture negative, while 19% died of the disease. The 72% culture negative rate by the end of six months is more than the 62% rate of culture conversion reported by Molla et al. among patients treated at treatment centres in the Amhara and Oromia regions of Ethiopia, but the 19% death rate is more than the 10% death rate reported by the same authors [38]. A substantial percentage (31%) of the patients had co-morbidity with MDR-TB, 83% of which was due to HIV. A study conducted in South Africa cited that, compared with HIV-negative patients, HIV-positive patients with MDR-TB had a lower chance of culture conversion and a higher chance of death [49]. In this view, the higher proportion of MDR-TB and HIV co-infection rate among patients included in this study might be a risk factor for the observed higher proportion of death among patients included in this study. It is worth noting that 87% (26/30) of the total deaths revealed in this study occurred during the first six months after commencing treatment, while only 13% of the total deaths occurred during the subsequent 18 months of treatment. This finding signals the need for intensive care during the initial months of patient treatment for MDR-TB.

The 69% composite treatment success rate revealed in this study is less and the 27% death rate is higher respectively than the 75% treatment success rate and 15% death rate reported by Molla et al. [38]. The 69% treatment success rate is also lower than the 78.6% treatments success rate reported by Meressa et al. [39]. However, the result is similar to the 70.6% treatment success rate reported by Anderson et al. [42], According to the report of Anderson et al., HIV co-infection with MDR-TB is associated with a higher rate of death from MDR-TB. In this view, the relatively low treatment success rate compared to the report by Meressa et al. [39], might be due to the higher rate of MDR-TB and HIV/AIDS co-infection observed among patients included in this study. Total of 46 (35%), (n = 132), patients were not employed and 70 (53%) were employed in the informal sector, mainly in daily labour works. This indicates the low socio-economic status of the patients included in this study. It is repeatedly cited in the literature that low monthly household income, living in poverty and unemployment are predictors of poor treatment outcome among patients with MDR-TB [9, 40]. In this view, the high proportion of unemployment among patients included in this study is a potential

challenge for patients to adhere to the standard schedule of the treatment given for MDR-TB. Most patients included in this study experienced at least one episode of adverse drug reactions in the course of their treatment. According to the reports by the World Health Organisation (WHO) [41] the presence of some co-morbidity with MDR-TB increases the risk of occurrence of adverse drug reactions from second-line drugs. In the United Kingdom [42] and Nigeria [43], it was documented that the presence of any co-morbidity with MDR-TB is associated with poor treatment outcome and high mortality among patients treated for MDR-TB. Caminero (2013) also cited that for the poor patients with MDR-TB, malnutrition impairs recovery [30]. Another report by the WHO [44] indicated that low body mass index (BMI) and lack of adequate weight gain during treatment are associated with death and increases the chance of occurrence of adverse drug reactions. Similarly, the report by Yuan et al. [45]; Vishakha et al. [46] and that of Lange et al. [47] each indicated that malnutrition is a risk factor for low cure rate and high rate of death among the poor patients with MDR-TB. In this study, the odds of death was significantly higher among patients with BMI less than $18.5 kg/m^2$, (AOR = 2.734, 95%CI: 1.01–7.395; P<0.048) and those with some co-morbidity at baseline (AOR = 4.260, 95%CI: 1.607–11.29; p<0.004). In view of extant literature, the high level of co-morbidity and malnutrition associated with MDR-TB revealed in this study could be a risk factor for the observed high proportion of death and high prevalence of adverse drug reactions among patients included in this study.

The result of this study indicates that addressing co-morbidities and MDR-TB associated malnutrition is key to improving treatment outcomes of patients treated for MDR-TB. The absence of optimum respiratory MDR-TB infection control practice at patients' household level and the use of public transport services by patients, revealed in this study, are potential risk factors for disease transmission among the community. Moreover, community level respiratory MDR-TB infection control effort did not meet the respiratory MDR-TB infection control recommendations of the Federal Ministry of Health of Ethiopia [16]. The observed 6% MDR-TB cases diagnosed among close contacts is more than the 3% to 5.4% reported from Peru [48]. According to the report of Scardigli et al. [49], poor respiratory MDR-TB infection control increases the risk of MDR-TB transmission to close contacts. In high HIV-prevalent settings, the situation amplifies disease occurrence within families. Thus, the poor MDR-TB infection control and the high prevalence of HIV among patients with MDR-TB, revealed in this study, are potential risk factors for an increase in the number of patients with MDR-TB in the community. The 6% prevalence of MDR-TB among close contacts is a warning sign regarding community and household level respiratory MDR-TB infection control in the study areas.

## Conclusion

If the problem of MDR-TB and the factors determining the treatment outcomes of patients with MDR-TB are to be tackled successfully, the factors determining the treatment outcomes of patients with MDR-TB need to be identified. In this regard, this study has identified socio-demographic and clinical factors that determine the treatment outcomes of patients with MDR-TB. Thus, the results from this study will enable health decision makers and caregivers for MDR-TB in Ethiopia to make evidence-informed decisions regarding the MDR-TB programme design and its management and resource allocation decisions during the subsequent national effort to expand the programmatic management of MDR-TB in the country.

## Recommendations

Improving socio-economic and nutritional support and provision of integrated care on MDR-TB and HIV/AIDS is recommended to mitigate the higher level of death among patients

treated for MDR-TB. To mitigate the inadvertent transmission of MDR-TB among the community, due emphasis should be given to respiratory MDR-TB infection control efforts. Lastly, the authors recommend further study to investigate factors behind the gender-based differentials of MDR-TB treatment outcomes revealed in this study.

## Supporting information

**S1 File.**
(DOCX)

**S1 Data.**
(SAV)

## Acknowledgments

We express our deep gratitude for the University of South Africa and Oromia Region Health Bureau, Department of Public Health Emergency Management and Health Research for providing us with ethical clearance required to conduct the study. We are also thankful to the management and the caregivers at Adama and Nekemte hospitals for assisting us with accessing data needed to conduct the study.

## Author Contributions

**Conceptualization:** Mengistu K. Wakjira.

**Data curation:** Mengistu K. Wakjira.

**Formal analysis:** Mengistu K. Wakjira.

**Funding acquisition:** Mengistu K. Wakjira.

**Investigation:** Mengistu K. Wakjira.

**Methodology:** Mengistu K. Wakjira.

**Project administration:** Mengistu K. Wakjira, Peter T. Sandy.

**Resources:** Mengistu K. Wakjira.

**Software:** Mengistu K. Wakjira.

**Supervision:** Mengistu K. Wakjira, Peter T. Sandy, A. H. Mavhandu-Mudzusi.

**Validation:** Mengistu K. Wakjira, Peter T. Sandy, A. H. Mavhandu-Mudzusi.

**Visualization:** Mengistu K. Wakjira.

**Writing – original draft:** Mengistu K. Wakjira.

**Writing – review & editing:** Mengistu K. Wakjira.

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
