## [Decision Letter · Decision Letter 0]

26 May 2021

PONE-D-20-39516

TREATMENT OUTCOMES OF PATIENTS WITH MDR-TB AND ITS DETERMINANTS AT REFERRAL HOSPITALS IN ETHIOPIA

PLOS ONE

Dear Dr. WAKJIRA,

Thank you for submitting your manuscript to PLOS ONE. After careful consideration, we feel that it has merit but does not fully meet PLOS ONE’s publication criteria as it currently stands. Therefore, we invite you to submit a revised version of the manuscript that addresses the points raised during the review process.

As suggested by reviewers, the manuscript needs to be rewritten. The text has to be shortened and made more crisp and clear.There are some additional comments:

1. The period of study is from 26th Dec, 2012 to 17th Sep,2017. During this period a total of 182 patients with MDR-TB were enrolled to treatment of which 46 patients did not meet the inclusion criteria. The inclusion and exclusion criteria should be specified.

2. The data was collected from 10th Nov,2016 to 7th Feb 2017. How was the 24 months follow up assessed for patients enrolled in Sep, 2017?

3. Abstract- Background- 3rd line may be modified to 'Thus, this study investigated the determinants responsible for poor treatment outcome in patients with MDR-TB at referral hospitals in Ethiopia.

4. Abstract- Results-4th line- Delete 'were' after 27%, also elsewhere also in the manuscript

5. Materials and Methods-Findings- General characteristics of patients- Employment status is given for 132/136 patients although there is heading 'other'

6. Clinical characteristics of the participants-'135 (99%) of the total patients had a drug-susceptibility test result for rifampicin and were resistant to it. Only fifty-eight (43%) of the patients had drug susceptibility

test result for both rifampicin and isoniazid and were resistant to both drugs'- How was DST performed? Was RR determined by GeneXpert? If so why only 58(43%) samples tested against both drugs?

7. Table 3- Please correct whether a total of 30 patients died or 34 died? Similarly, comorbidity has been shown as 41/136 in Table 1 and 34 (favourable+unfavourable outcome). Also, HIV positive patients are 34 in Table 1 and 27 in Table 3.

8. Discussion-'...it was documented that presence of any comorbidity with MDR-TB is associated with poor treatment outcome and high mortality among patients treated for drug-resistant tuberculosis. In this view, the high level of comorbidity with MDR-TB (31%) revealed in this study could be a risk factor for the observed high prevalence of adverse drug-reactions higher proportion of death among patients included in this study.'- '...64% prevalence of malnutrition among patients included in this study might explain the significantly higher rate of death and high prevalence of adverse drug reactions observed among patients included in this

study.

The association of comorbidity and malnutrition for adverse drug reaction seems inappropriate unless suitable references can be cited.

We look forward to receiving your revised manuscript.

Kind regards,

Shampa Anupurba, MD

Academic Editor

PLOS ONE

Journal Requirements:

Furthermore, please provide additional information in the methods section regarding the content, development and validation of the questionnaire used in the study.

Please provide a justification for the sample size used in your study, including any relevant power calculations (if applicable).

3. Please specify what type you obtained (for instance, written or verbal). If your study included minors under age 18, state whether you obtained consent from parents or guardians. If the need for consent was waived by the ethics committee, please include this information.

4. We suggest you thoroughly copyedit your manuscript for language usage, spelling, and grammar. If you do not know anyone who can help you do this, you may wish to consider employing a professional scientific editing service. 

6. Thank you for stating the following in the Competing Interests section:

We note that one or more of the authors are employed by a commercial company: Abt Associates Inc Ethiopia.

7. Please respond by return e-mail with an updated version of your manuscript to amend either the abstract on the online submission form or the abstract in the manuscript so that they are identical. We can make any changes on your behalf.

Reviewers' comments:

Reviewer's Responses to Questions

**Comments to the Author**

1. Is the manuscript technically sound, and do the data support the conclusions?

Reviewer #1: Partly

Reviewer #2: Partly

2. Has the statistical analysis been performed appropriately and rigorously? 

Reviewer #1: Yes

Reviewer #2: No

3. Have the authors made all data underlying the findings in their manuscript fully available?

Reviewer #1: Yes

Reviewer #2: No

4. Is the manuscript presented in an intelligible fashion and written in standard English?

Reviewer #1: Yes

Reviewer #2: No

5. Review Comments to the Author

Reviewer #1: 1. The study topic is relevant.

2.The Abstract is well written.

3. Aim & Objectives are clear.

Materials & Methods: The study design needs clarification about sample size, how the sample size was calculated.

Duration of the study is confusing as in section "study setting & sampling" it is mentioned as between 26th of Dec 2012 to 17th of Sept. 2016. however in section "data collection & analysis" the duration is different. This should be clarified.

The inclusion and exclusion criteria has not been given. How the data was collected, details should be included.

Table 1 HIV/ AIDS mentioned at two places .

Table2. Culture Positive 0% , written twice. If 27 cases died within 6months total cases remained are 109 not 110. hence this should be clarified; and in 24 months period 30 cases died. so total cases remain 109-30=79, The calculation is not clear,

Table 3 & table 4: overlapping data should be removed.

References: The page No. are missing in few articles.

Reviewer #2: 1. The study is based on data from 2012 till September 2016. It is a relatively old data. To make the study more relevant in the present scenario, the authors are advised to include data at least till December 2020.

2. There is discrepancy in data throughout the text, e.g., at one place it is mentioned there are total 30 deaths, at another place it is mentioned it includes 19 males and 15 females that amounts to 34. In table also it is mentioned as 19 and 15. The article should be read thoroughly and the statistical errors need to be resolved.

3. The ‘Structural Questionnaire’ that was used to collect data should have been shared as an appendix/supplement.

4. The study design is not clear. In one sentence it is written-‘both deductive and inductive approaches were used’. However, in the next sentence it is written- ‘Only deductive approach was used’. This needs clarification.

5. There is non-uniformity in writing numerical values-at some place in text, it is written as digits and at some places as words.

6. What drug regimen was provided to MDR-TB patients? Is it the same regimen prescribed to all patients or it is different for pulmonary and extrapulmonary cases and also in the same group? If different regimes are given to different patients, then it would be a major confounding factor in analyzing the disease outcome.

7. Does the patient need to pay for the treatment of TB or the drugs are provided free of cost by government of Ethiopia?

8. The reference for classification of initial bacillary load as ‘scanty, moderate or high’ is required. It needs clear objective definition.

9. Which drug was responsible for suicide due to clinical psychiatric problem as an adverse drug reaction? How the causal relationship was established?

10. The analysis should also be done comparing urban and rural population and its correlation with disease outcome.

11. The English language of the article needs to be corrected and improved.

12. There was no new information in the conclusion of the study. These are already established facts that malnutrition, poor socio-economic status, unemployment, co-infection with HIV and co-morbidities are responsible for poor disease outcome in MDR-TB cases.

6. PLOS authors have the option to publish the peer review history of their article (what does this mean?). If published, this will include your full peer review and any attached files.

Reviewer #1: No

Reviewer #2: No

---

## [Author Response · Author response to Decision Letter 0]

15 Oct 2021

Please make note of the funding statement I included in the revised cover letter to update it online on behalf of me! Funding Statement: During the time the research was conducted one of the authors was employed to Abt Associates Inc. operating in Ethiopia. “The funder provided support in the form of salaries for authors (MK Wakjira), but did not have any additional role in the study design, data collection and analysis, decision to publish, or preparation of the manuscript. The specific roles of these authors are articulated in the ‘authors contributions section.’”

---

## [Decision Letter · Decision Letter 1]

23 Dec 2021

TREATMENT OUTCOMES OF PATIENTS WITH MDR-TB AND ITS DETERMINANTS AT REFERRAL HOSPITALS IN ETHIOPIA

PONE-D-20-39516R1

Dear Dr. WAKJIRA,

We’re pleased to inform you that your manuscript has been judged scientifically suitable for publication and will be formally accepted for publication once it meets all outstanding technical requirements.

Kind regards,

Shampa Anupurba, MD

Academic Editor

PLOS ONE

Additional Editor Comments (optional):

Reviewers' comments:

Reviewer's Responses to Questions

**Comments to the Author**

1. If the authors have adequately addressed your comments raised in a previous round of review and you feel that this manuscript is now acceptable for publication, you may indicate that here to bypass the “Comments to the Author” section, enter your conflict of interest statement in the “Confidential to Editor” section, and submit your "Accept" recommendation.

Reviewer #2: All comments have been addressed

2. Is the manuscript technically sound, and do the data support the conclusions?

Reviewer #2: Yes

3. Has the statistical analysis been performed appropriately and rigorously? 

Reviewer #2: Yes

4. Have the authors made all data underlying the findings in their manuscript fully available?

Reviewer #2: Yes

5. Is the manuscript presented in an intelligible fashion and written in standard English?

Reviewer #2: Yes

6. Review Comments to the Author

Reviewer #2: This is a re-review of the manuscript. I have read all the answers of the authors and their justification for being unable to incorporate some of the suggestions. The authors have appropriately answered and incorporated comments made by the reveiwer.

7. PLOS authors have the option to publish the peer review history of their article (what does this mean?). If published, this will include your full peer review and any attached files.

Reviewer #2: No

---

## [Editor Report · Acceptance letter]

9 Feb 2022

PONE-D-20-39516R1 

Treatment outcomes of patients with MDR-TB and its determinants at referral hospitals in Ethiopia 

Dear Dr. Wakjira:

I'm pleased to inform you that your manuscript has been deemed suitable for publication in PLOS ONE. Congratulations! Your manuscript is now with our production department. 

Kind regards, 

on behalf of

Dr. Shampa Anupurba 

Academic Editor

PLOS ONE